# Genome-wide association and expression quantitative trait loci in cattle reveals common genes regulating mammalian fertility

Mehrnush Forutan [1] ✉, Bailey N. Engle [1,2], Amanda J. Chamberlain [3,4], Elizabeth M. Ross[1], Loan T. Nguyen[1], Michael J. D'Occhio[5], Alf Collins Snr[6], Elise A. Kho[1], Geoffry Fordyce[1], Shannon Speight[7], Michael E. Goddard[3,8] & Ben J. Hayes [1]

Most genetic variants associated with fertility in mammals fall in non-coding regions of the genome and it is unclear how these variants affect fertility. Here we use genome-wide association summary statistics for Heifer puberty (pubertal or not at 600 days) from 27,707 *Bos indicus*, *Bos taurus* and crossbred cattle; multi-trait GWAS signals from 2119 indicine cattle for four fertility traits, including days to calving, age at first calving, pregnancy status, and foetus age in weeks (assessed by rectal palpation of the foetus); and expression quantitative trait locus for whole blood from 489 indicine cattle, to identify 87 putatively functional genes affecting cattle fertility. Our analysis reveals a significant overlap between the set of cattle and previously reported human fertility-related genes, impling the existence of a shared pool of genes that regulate fertility in mammals. These findings are crucial for developing approaches to improve fertility in cattle and potentially other mammals.

Fertility in mammals is a complex trait, affected by many loci and environmental variation[1]. In both beef and dairy cattle, female fertility is a key trait associated with productivity[2]. Identifying mutations underpinning variation in fertility between individuals has been challenging, as the large environmental variation typical of fertility traits[3] means extremely large sample sizes are required. Notwithstanding, progress has been made, and several genome-wide association studies (GWAS) and quantitative trait loci (QTL) mapping studies in cattle and human populations reported genomic regions affecting female fertility traits[4,5]; In cattle, multiple significant GWAS signals were reported in a number of studies[4,6], however, the overlap of QTL locations among different populations is generally poor[7], demonstrating the challenges in finding candidate genes and mapping causal variants for cattle fertility.

The SNPs on standard SNP arrays are preselected to be highly polymorphic across breeds[8] and, typically do not include causal variants. Array SNP may, therefore, not be in strong linkage disequilbrium (LD) with rare or breed-specific causal variants, therefore the effects of these variants may not be captured in GWAS[8]. In contrast to SNP arrays, using whole genome sequence data could improve the power of GWAS since the causal variants should be included in the sequence data[9]. However, even using whole genome sequence data, the identification of the causal variants for a complex trait remains difficult, due to the small effect size of most causal variants and LD between variants[10]. Particularly in cattle, there are typically many variants in high LD, any one of which could be the cause of the variation in phenotype[9].

Causal variants are often pleiotropic, i.e. affecting more than one trait, so multi-trait analyses may result in greater power to detect QTL and more precise mapping[11,12]. For example, the conditional multi-trait (CMT) GWAS considers the estimated effects of the variants as well as the *p*-value and direction of effect for each trait[8].

GWAS only identifies SNPs strongly associated with the trait of interest, without revealing the underlying biological mechanism. Using an expression quantitative trait loci (eQTL), where gene expression levels are modeled as a phenotype, may identify mutations affecting complex trait

[1]Queensland Alliance for Agriculture and Food Innovation, The University of Queensland, St Lucia, QLD, Australia. [2]USDA,ARS, U.S. Meat Animal Research Center, Clay Center, NE 68933, USA. [3]Agriculture Victoria, Centre for AgriBiosciences, Bundoora, VIC, Australia. [4]School of Applied Systems Biology, La Trobe University, Bundoora, VIC 3083, Australia. [5]School of Life and Environmental Sciences, Faculty of Science, The University of Sydney, Sydney, NSW, Australia. [6]Collins Belah Valley Brahman Stud, Marlborough 4705 QLD, Australia. [7]Black Box Co, Mareeba, QLD, Australia. [8]University of Melbourne, Melbourne, Australia. ✉e-mail: m.forutan@uq.edu.au

variation via the regulation of gene expression[13]. Combining eQTL and complex trait GWAS information could be a pathway to identify causative mutations more precisely[14].

This study was carried out to (1) identify the genetic variants and genes associated with four fertility-related traits in a well-phenotyped cattle population, and (2) investigate the overlap between the potential cattle and human fertility-related genes. We first identified significant variants and genes associated with four fertility-related traits including days to calving (DTC), age at first calving (AFC), pregnancy status (preg_st), and foetal age in weeks (wks_preg) (Fig. 1) by performing a stepwise conditional multi-trait GWAS (CMT-GWAS) analysis in a well-phenotyped *Bos indicus* (indicine) cattle population with imputed whole genome sequence data (discovery population). The results were validated in a cohort of 28k indicine, *Bos taurus* (taurine) and crossbred cattle (validation population) recorded for Heifer puberty[15]. We also identified eQTLs for more than 10,000 genes expressed in whole blood of 489 indicine cattle from the same population (discovery population). We then identified genes whose expression levels were associated with fertility due to pleiotropy or causality by integrating GWAS results with eQTL data using summary-data-based mendelian randomisation (SMR) analysis. To explore the connection between fertility traits and functional genomic regions, we explored the overlap of the lead eQTLs with reported ChIP-seq and ATAC-seq peaks identified across publicly available tissues in different studies[16,17]. We identified 12 genes that are common between the set of potential cattle fertility related genes and genes reported to affect age at natural menopause and menarche in humans[1,18], suggesting that a set of common genes regulates fertility in mammals.

## Results and Discussion

### Single and multi-trait GWAS results in indicine cattle (discovery population)

Single trait genome-wide association studies for four heifer fertility traits (DTC, AFC, preg_st, wks_preg) were performed using phenotypes and imputed genotypes (31 million whole genome sequence (WGS) variants) from 2,119 indicine cattle (discovery population) (Fig. 1; Supplementary Data 1). For all fertility traits, there was no indication of inflation of the test statistic due to population structure (Fig. 2a).

In the single trait GWAS, the greatest number of associated SNPs were detected for AFC (Supplementary Data 2). There were 59, 3, 4, and 18 significant variants at a threshold of $P < 5 \times 0^{-8}$ for AFC, DTC, preg_st, and wks_preg, respectively (Fig. 2b), in 21 clusters across the genome. This corresponds to the false discovery rate (FDR) ranging from 0.51 (DTC) to 0.02 (AFC). The significant variants associated with AFC ($P < 5 \times 10^{-8}$) clustered on chromosomes 2, 6, 14, 15, 16, 17, 18, and 19 (Supplementary Data 2). The significant variants ($P < 5 \times 10^{-8}$) for DTC were on 14 and 21, preg_st on 7, 8 and 19, and wks_preg on 5, 6 and 14 (Supplementary Data 2). Some of these were close to or had overlaps with QTLs identified in previous GWAS studies. For example, we detected two variants on Chr 19 associated with preg_st at 12.1 Mb, which were close to a missense mutation (rs383232842, p.H210R) in *TUBD1*, previously detected for stillbirth in Braunvieh and Fleckvieh cattle[19]. Consistent with our results, another study in Nordic Red and Danish Jersey cattle[20] reported a haplotype at 10,920,596-11,863,651 bp on Chr 19 associated with non-return rate at 56, 100, and 150 days. In the current study, a QTL was also identified at 86 Mb on Chr 6 for wks_preg, which was close to the QTL at 85.8 Mb previously detected for the interval between first and last insemination, a trait that is used for the

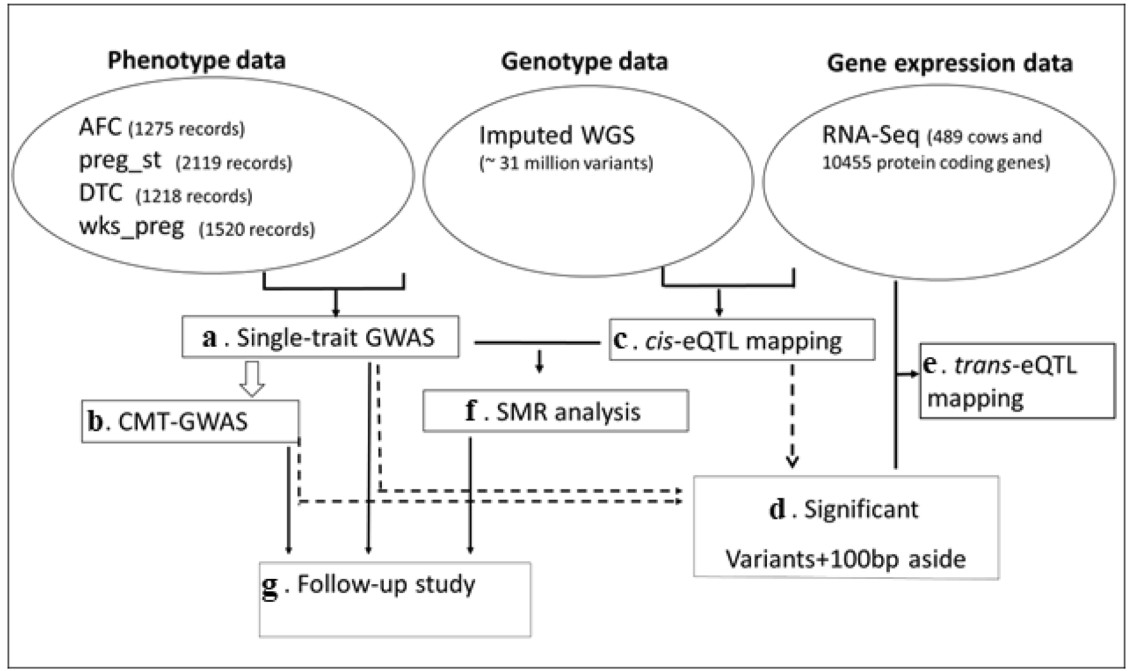

**Fig. 1 | Flow chart of the analysis conducted in the discovery population (Bos indicus cattle).** Analysis was performed in order a to g as follows: **a** Single trait genome-wide association studies for four fertility traits (DTC, AFC, preg_st, wks_preg) were performed using 31 million imputed whole genome sequence (WGS) variants. **b** A stepwise conditional multi-trait GWAS (CMT-GWAS) analysis was performed for fertility traits (DTC, AFC, preg_st, wks_preg). **c** Cis-eQTL discovery was performed for a sample of 489 heifers and cows with RNA-seq and imputed WGS data. **d** The most significant cis-eQTLs for each eGene (n = 4376, FDR < 0.01) were selected. Additionally, we added 225 genome-wide significant GWAS variants from the single trait and CMT-GWAS model described above, applying a significance threshold $P \le 5 \times 10^{-8}$. We also added variants within 100 bp either side of each selected variant. **e** Trans-eQTL mapping for significant variants identified through cis-eQTL or GWAS analysis. We specifically targeted genes where the variant was located more than 5 Mb away from the gene on the same chromosome. **f** Summary-data-based mendelian randomisation (SMR) analysis was performed by integrating GWAS with cis-eQTL results. **g** Follow-up study of the final combined list of significant genes associated with fertility in cattle.

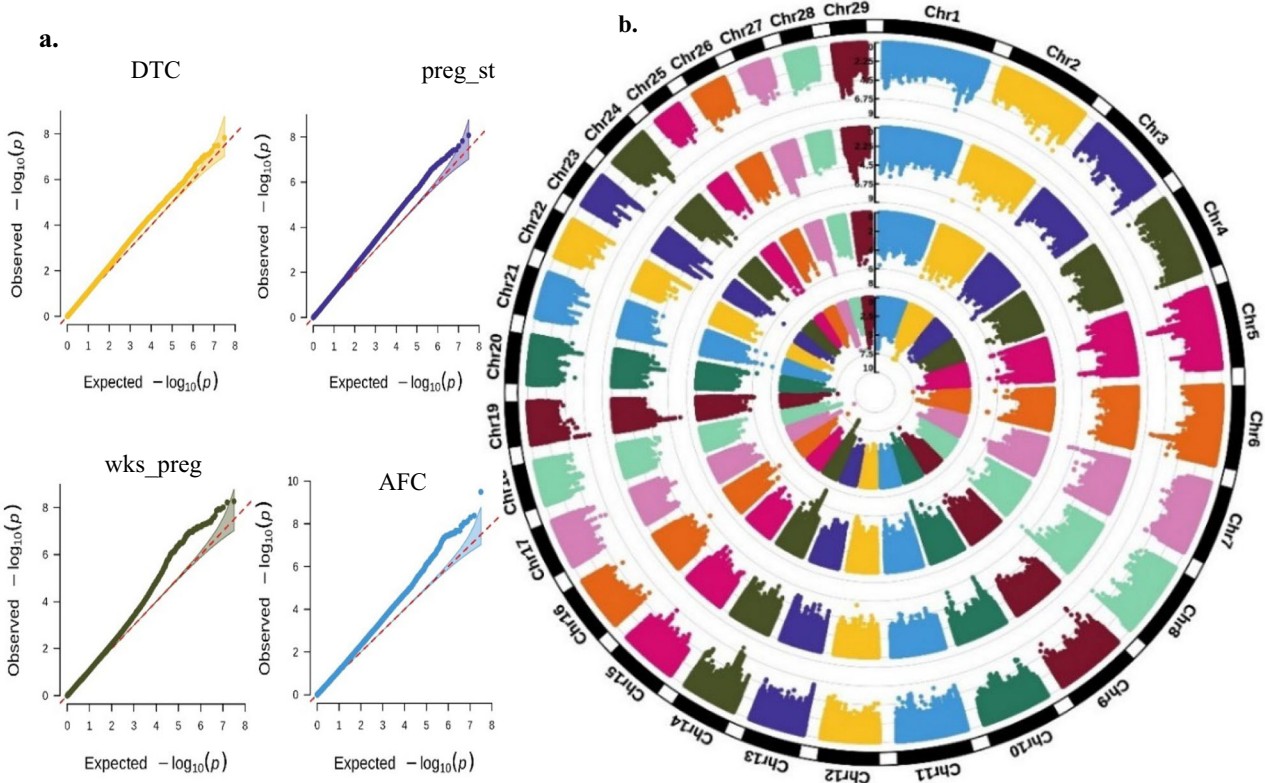

**Fig. 2 | GWAS analysis of fertility traits in *Bos indicus* cattle (discovery population). a** Quantile-quantile (QQ) plot of the single-trait GWAS shown in **b** the Manhattan plot of single-trait GWAS analysis for four fertility traits, from inner most to outer most, heifer age of calving (AFC), heifer days to calving (DTC), heifer pregnancy success (preg_st), and heifer weeks pregnant (wks_preg).

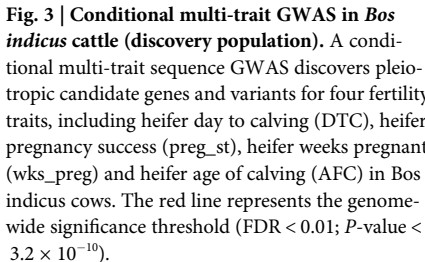
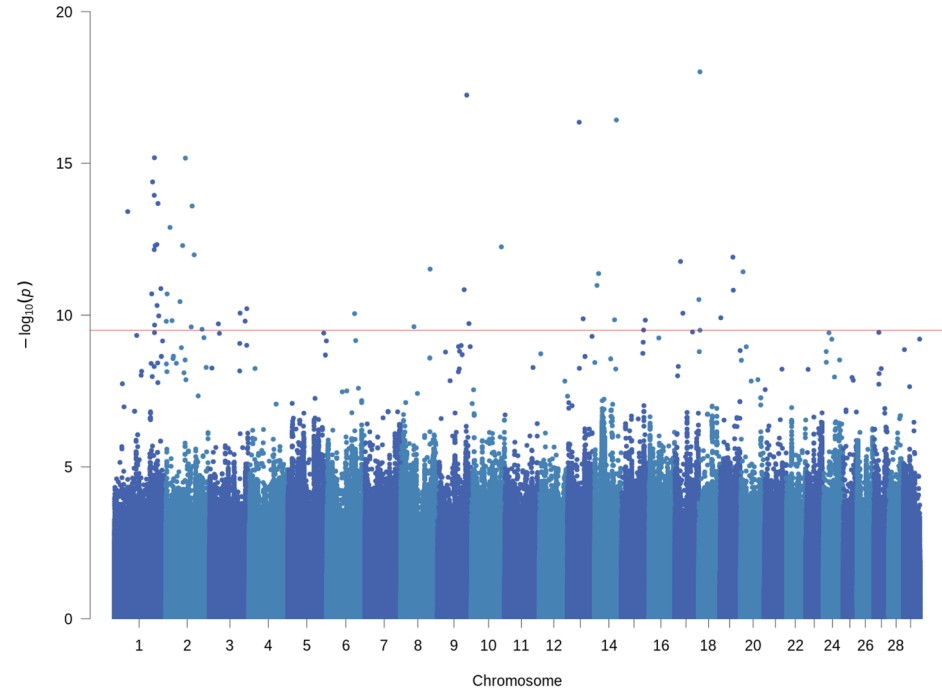

**Fig. 3 | Conditional multi-trait GWAS in *Bos indicus* cattle (discovery population).** A conditional multi-trait sequence GWAS discovers pleiotropic candidate genes and variants for four fertility traits, including heifer day to calving (DTC), heifer pregnancy success (preg_st), heifer weeks pregnant (wks_preg) and heifer age of calving (AFC) in Bos indicus cows. The red line represents the genome-wide significance threshold (FDR < 0.01; *P*-value < $3.2 \times 10^{-10}$).

capacity of cows to conceive and maintain a pregnancy in both Chinese and Nordic Holsteins[4].

The CMT-GWAS (multitrait) method resulted in more highly significant peaks and a lower FDR than any of the single-trait GWAS (FDR < 0.01; $P < 3.2 \times 10^{-10}$) (Fig. 3). The genes closest to these peaks are presented in Supplementary Data 3. The most significant putative causal variant identified by CMT-GWAS was an intergenic variant Chr18:3753117 ($P = 9.6 \times 10^{-19}$) located 47 kb from the contactin-associated protein family member 4 gene (*CNTNAP4*) and 88 kb from the *LOC112442251*. *CNTNAP4* is expressed in various regions of the brain, including the

developing cortical interneurons and midbrain dopaminergic neurons in mice[21]. In goats, *CNTNAP4* has been identified as one of the most promising novel candidate genes for reproductive performance which could be used as a marker of reproduction and fertility[22]. The second most significant variant was Chr14:67,194,407 ($P = 3.7 \times 10^{-17}$) located in an intron of the gene carboxypeptidase Q *(CPQ)*, and 340 kb from Testis-specific protein Y-encoded-like 5 *(TSPYL5)*. In some genome-wide association studies *TSPYL5* was associated with plasma oestradiol levels in humans (e.g., ref. 23). Another putative causal variant (Chr2:61024817; $P = 6.7 \times 10^{-16}$) was located 225 kb from Chemokine receptor type 4 *(CXCR4)*. *CXCR4* expression is a potential biomarker to predict implantation competence of an embryo in human in vito fertilization[24]. It has also been implicated in the ovulatory process in equine and bovine[25].

## Validation results in indicine, taurine and crossbred cattle (validation population)

Of the 225 significant GWAS variants ($P < 5 \times 10^{-8}$) identified from the individual single-trait and conditional multi-trait GWAS analyses in the (indicine) discovery population, 11 variants ($P < 0.002$; FDR = 3.8%; Supplementary Data 4) on Chr 5 (8 between 48,005,346 and 48,013,239 bp), and Chr 14 (23,436,596 : 23,442,688 and 61,428,259 bp), were validated in the validation population. This was substantially more than expected by chance ($P < 0.05$). The validated, significant signal on Chr 5 is within 39 kb of the gene *HMGA2*, which has been identified as an oncogene and shown to be an upstream regulator of *PLAG1* expression, a gene that was reported to habour a mutation with significant effects on fertility[26,27]. The two significant variants near 23.4 Mb on Chr 14 are in an intron and upstream of the gene short-chain dehydrogenase/reductase family 16C member 5 *(SDR16C5)*, and about 60 kb from coiled-coil-helix-coiled-coil-helix domain containing 7 *(CHCHD7)* and *PLAG1*. Genes mapped on Chr 14, particularly *PLAG1* and *CHCHD7*, have been reported to be associated with blood concentrations of GH, IGF1, and IGF2 which were significantly associated with puberty in indicine heifers[6]. Another validated variant, Chr14:61,428,259, was located in an intron region of the gene ATPase H+ transporting V1 subunit C1 *(ATP6V1C1)*, a potential gene candidate that predicts oocyte competence and is used to select embryos suitable for transfer to a recipient[28].

We estimated the proportion of phenotypic variance accounted for by the 225 potential causal variants for indicine in the validation population (animals with indicine content equal or greater than 0.80, as the breed composition of these animals more closely matched the discovery population) and found that these loci explained 2.8% of the variation, which was more than the average variance explained by random subsets of 225 variants (1.7% ± 0.004). The relatively low validation rate might reflect the fact that different proxy traits were used for cattle fertility in the discovery and validation population. The probability of observing false associations between a specific trait and SNPs across different populations is low, especially when significant associations are verified across two populations and for different proxy traits. The current results provide strong evidence that SNPs for heifer puberty are concentrated at certain areas on Chr 5 and 14, and these areas warrant further study to identify causal variants for female fertility traits in beef cattle.

We also conducted a GWAS in the validation population, as the large size of this population allows for a powerful GWAS, albeit for a single trait (heifer puberty). This GWAS with 27,707 indicine, taurine and crossbred cattle with imputed sequence data revealed four gene regions that were highly associated with heifer puberty, including pleomorphic adenoma gene 1 *(PLAG1)*, high mobility group AT-hook 2 *(HMGA2)*, ligand dependent nuclear receptor corepressor-like *(LCORL)*/ non-SMC condensin I complex, subunit G *(NCAPG)* and serpin family B member 9 *(SERPIN9)* (Fig. 4a). Gene expression studies in indicine cattle[29] and pigs[30] have indicated that *PLAG1*, *HMGA2*, and *LCORL/NCAP* are either not expressed or expressed at significantly lower levels in adults compared to their expression levels in foetuses (e.g., Fig. 4b, c[31]; Table 1[29]). These genes, and eQTL for these genes, are inlikely to be identified in whole blood of mature animals.

Interestingly, alleles of *PLAG1* are associated with either early puberty and lower mature size, or alternatively, later puberty and taller stature[27,32,33]. Taken together, the effect of the alleles of *PLAG1* and the lack of expression of puberty-associated genes in heifers and cows from gene expression atlas data (Fig. 4b, c) suggest that much of the roadmap for allocation of resources to reproduction versus growth likely occurs very early in life, primarily during foetal development.

## *cis*-eQTLs and *trans*-eQTLs in indicine cattle

To discover eQTL, gene expression data was obtained from whole blood samples from 489 heifers and cows from the indicine discovery population (Supplementary Data 5). *Cis*-eQTL (SNP-gene distance < 2 Mb, default value in OSCA[34]), and *trans*-eQTL (SNP-gene distance >5 Mb on the same chromosome) analyses were performed using 10,455 autosomal genes from 489 cattle. We adopted a hierarchical multiple testing correction procedure to account for multiple testing, which usually has better calibrated FDR in contrast to pooled methods[35]. The reproducibility of our eQTL results was assessed by splitting the sample into two approximately equally sized samples and calculating the proportion of significant *cis*-eGenes (genes with significant *cis*-eQTL) in subsample1 observed in subsample2, and vice versa (Supplementary Fig. 1a, b). On average we observed 81% internal reproducibility of *cis*-eGenes between half-samples at global-FDR threshold 0.01. All lead *cis*-eQTL (global-FDR < 0.01) identified in one half-sample showed the same direction of effect size, with a correlation of 0.99 in the other half-sample (Supplementary Fig. 1c, d).

Using the entire dataset, we identified 4376 *cis*-eGenes (41.8% of the 10,455 tested autosomal genes; global-FDR < 0.01; Fig. 5a). The five most significant *cis*-eQTLs were detected on chromosomes 10, 22, 26, 11, and 12. (Supplementary Data 6). All five *cis*-eQTLs were also detected using the subsamples (global-FDR < 0.01).

The proportion of all tested genes that were *cis*-eGenes (*ePercent*) was 41.8%, which is within the range of reports from a previous study[36]. Liu et al.[36] reported a correlation of 0.85 between the *ePercent* and sample sizes across 23 distinct tissues in cattle. Generally, the *ePercent* might be influenced by the tissue, sample size, diversity of the breeds used for sampling, sequencing depth, and sex, and age of animals[36].

Seventy-eight percent of the lead *cis*-eQTLs were close to the respective gene start site (within 100 kb) (Fig. 5b), and the lead *cis*-eQTLs with largest effects had the smallest distance to the gene start site (within 50 kb for 58% of the top 25% lead *cis*-eQTL, Fig. 5c), in agreement with previous studies (e.g., ref. 37). Also, around 22% of lead *cis-eQTLs* (951 out of 4376) were located within the respective gene. Our results demonstrated 4 to 7.5 fold enrichment of lead *cis*-eQTLs among promoter, proximal, five'- untranslated region (UTR), and three'-UTR regions (based on the ARS-UCD1.2 genome, the refGene table) as compared with what would be expected by chance (Hypergeometric test; $P$-value < $6.3 \times 10^{-9}$; Fig. 5d; Supplementary Data 7). The enrichment level of lead *cis*-eQTLs within bovine ChIP–seq peaks[16,17] (Methods) ranged from 1.34 (H3K27me3) to 3.64 (H3K4me1). The smallest and largest number of *cis*-eQTL were identified for Chr 12 and 7, respectively (Fig. 6).

We identified 1105 *trans*-eQTLs and 401 *trans*-eGenes (genes with significant *trans*-eQTLs; global-FDR < 0.01) (Fig. 7a). Overall, most of the lead *trans*-eQTL (259 out of 401; 64%) were located within 5–7 Mb up- or downstream of gene start sites (Fig. 7b, c). In agreement with a previous study[37], *trans*-eQTL, for which the SNP was located distal to the gene (>5 Mb), had smaller effect sizes than *cis*-eQTL (*t* test; $P$-value < 0.001). The average of absolute effect size across eGenes with at least one *cis*- and *trans*-eQTL (397 eGenes) was 0.28 and 0.33 for *trans*-eQTLs and *cis*-eQTLs, respectively. Võsa et al.[37] suggested that *trans*-eQTL could be relevant for complex traits, compared to stronger *cis*-eQTL effects, because each *trans*-eQTL effect is less likely to be dampened by compensatory post-transcriptional buffering or removed from the population by negative selection. They also reported that *trans*-eQTLs can affect many genes and have a widespread impact on regulatory networks[37]. We observed 148 eQTLs having a trans effect on at least 2 genes contributing to 338 *trans*-

**Fig. 4 | GWAS analysis for heifer puberty and gene expression profile for _PLAG1_ and _HMGA2_. a.** Genome wide association with 48.8 million SNP for heifer puberty. Odd chromosomes are coloured blue, even chromosomes in red. The significance threshold on the y-axis is 8.3 (e.g. -log10($5 \times 10^{-8}$)). Only SNPs with significance > 3 (_P_-value < 0.0001) are shown. **b.** Gene expression (Fragments Per Kilobase of transcript per Million mapped reads) in foetal and adult tissues for _PLAG1_ and **c.** _HMGA2_, data from the bovine gene expression atlas (Fang et al.[31]).

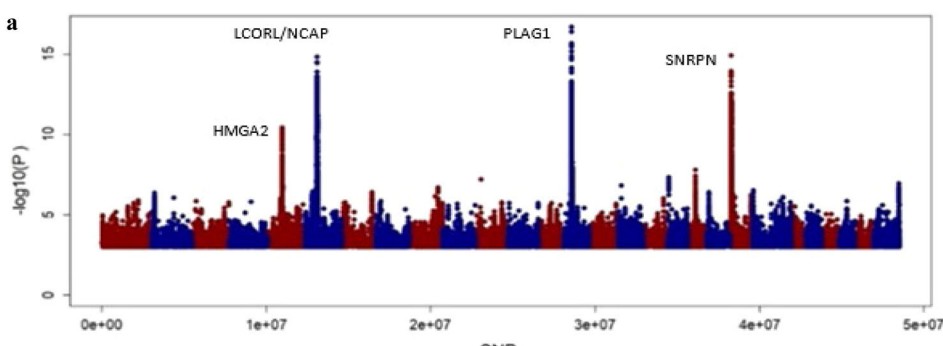

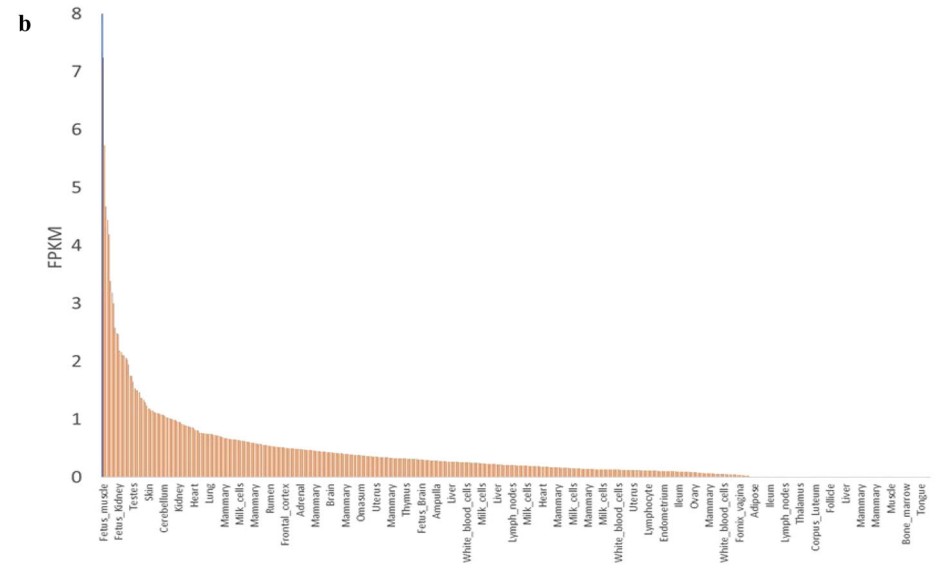

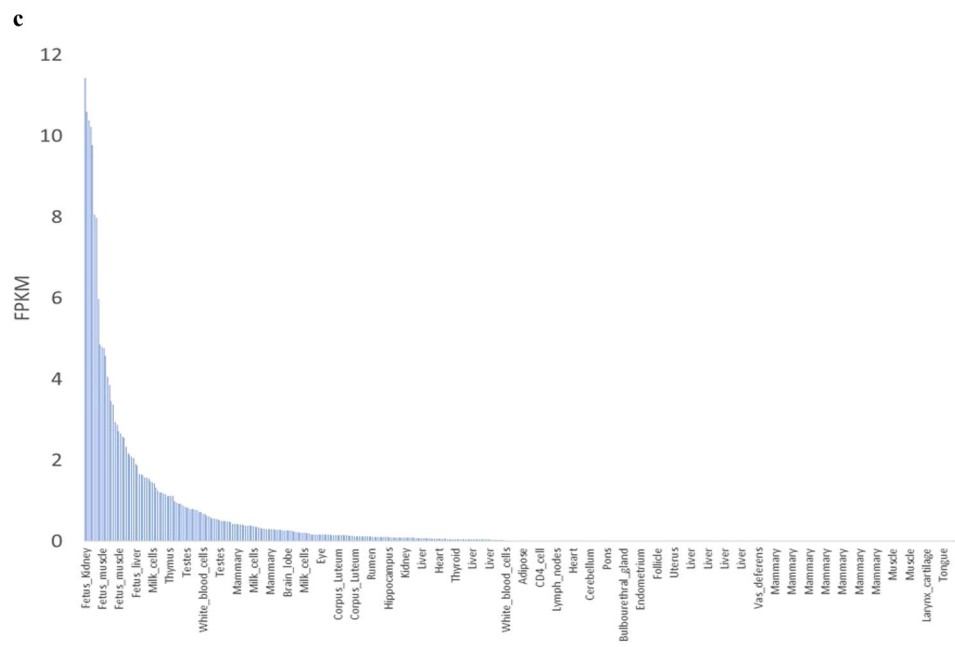

eQTLs. Interestingly, 755 eQTLs exhibited both cis and trans effects, with 448 of them exerting a cis effect on at least 2 genes, contributing to a total of 1870 _cis_-eQTLs.

In agreement with a previous study in cattle[32], the majority of lead _cis_-eQTL and _trans_-eQTL variants were non-coding variants (Fig. 5d and 7d; Supplementary Data 8). Thirteen (17.5) to 36 (31) percent of lead _trans_-eQTLs (_cis_-eQTLs) were located within bovine ChIP–seq and ATAC-seq peaks from different assays and tissues reported in different studies[16,17], in all cases more than expected by chance (Hypergeometric test; _P-value_ < $9.3 \times 10^{-6}$). For lead _trans_-eQTLs, the enrichment level ranged from 1.52 (ATAC-seq[38]) to 2.95 (CTCF binding site). This result indicates that the lead _trans_-eQTLs and _cis_-eQTLs were more likely to be in enhancer,

repressor, or promoter regions of genes, therefore these SNPs are likely to be involved in altering gene expression by changing the binding site sequences of transcription factors or other regulatory proteins. The datasets derived from ATAC-Seq and ChIP-seq experiments serve as valuable resources; by integrating these datasets with eQTL analyses, we can establish connections between genetic variants and functional genomic regions as well as pathways associated with complex traits.

## Integration of GWAS and eQTL results

Previous studies have suggested that significant GWAS loci which were also eQTLs were more likely to be causal[39]. To identify the mutations that affect both gene expression and fertility traits due to either pleiotropy or causality, we performed a summary data-based mendelian randomisation (SMR) test[40]. We identified eight genes that showed associations with fertility in the SMR test (Bonferroni corrected $P\text{-value}_{SMR} < 3.5 \times 10^{-4}$; 0.05/141 genes tested by the SMR analyses; Supplementary Data 9), suggesting that SNP effects on fertility traits could be mediated by genetic regulation of expression levels of these genes in whole blood.

One such variant, Chr5:47593069 T>C associated ($P\text{-value} < 1.7 \times 10^{-45}$) with expression of *HELB* (DNA helicase B), is an interesting candidate. *HELB* encodes a DNA-dependent ATPase which catalyses the unwinding of DNA necessary for DNA replication, repair, recombination, and transcription[5]. It has a substantial role in DNA damage repair in reproductive ageing[5]. A previous study identified nine SNPs in coding regions that are fixed for different alleles between taurine and indicine beef populations that affect the coding sequence of HELB[41]. SNPs were located in a 430 kb selective sweep on Chr 5 (47,438,392–47,865,772 bp) and span several genes including *HELB*, *IRAK3*, *TMBIM4*, *GRIP1*, and part of *HMGA2*. The authors suggested this sweep has been a result of selective breeding for improved adaptation of cattle to tropical conditions[41]. *HELB* has also been previously associated with puberty in both male and female tropically adapted cattle[42]. In

### Table 1 | Counts of RNA reads aligning to key genes from foetal and adult tissues, in samples from a Brahman Cow and her foetus (for details see ref. 29)

|  | Foetal liver | Foetal lung | Adult Liver | Adult Lung |
|---|---|---|---|---|
| IGF2 | 364,487 | 994,351 | 145,051 | 12,144 |
| IGF2R | 47,959 | 42,565 | 10,490 | 13,694 |
| IGF2BP3 | 3663 | 2037 | 35 | 95 |
| ZNF462 | 247 | 4934 | 85 | 2172 |
| BMP4 | 601 | 5947 | 306 | 2517 |
| LOC101905775 | 0 | 0 | 0 | 0 |
| ASIP | 16 | 22 | 13 | 17 |
| ZP3 | 2 | 7 | 2 | 10 |
| PLAG1 | 343 | 673 | 78 | 273 |
| HMGA2 | 269 | 926 | 0 | 3 |
| LCORL | 827 | 383 | 542 | 1071 |
| SNRPN | 1489 | 2911 | 1246 | 1545 |

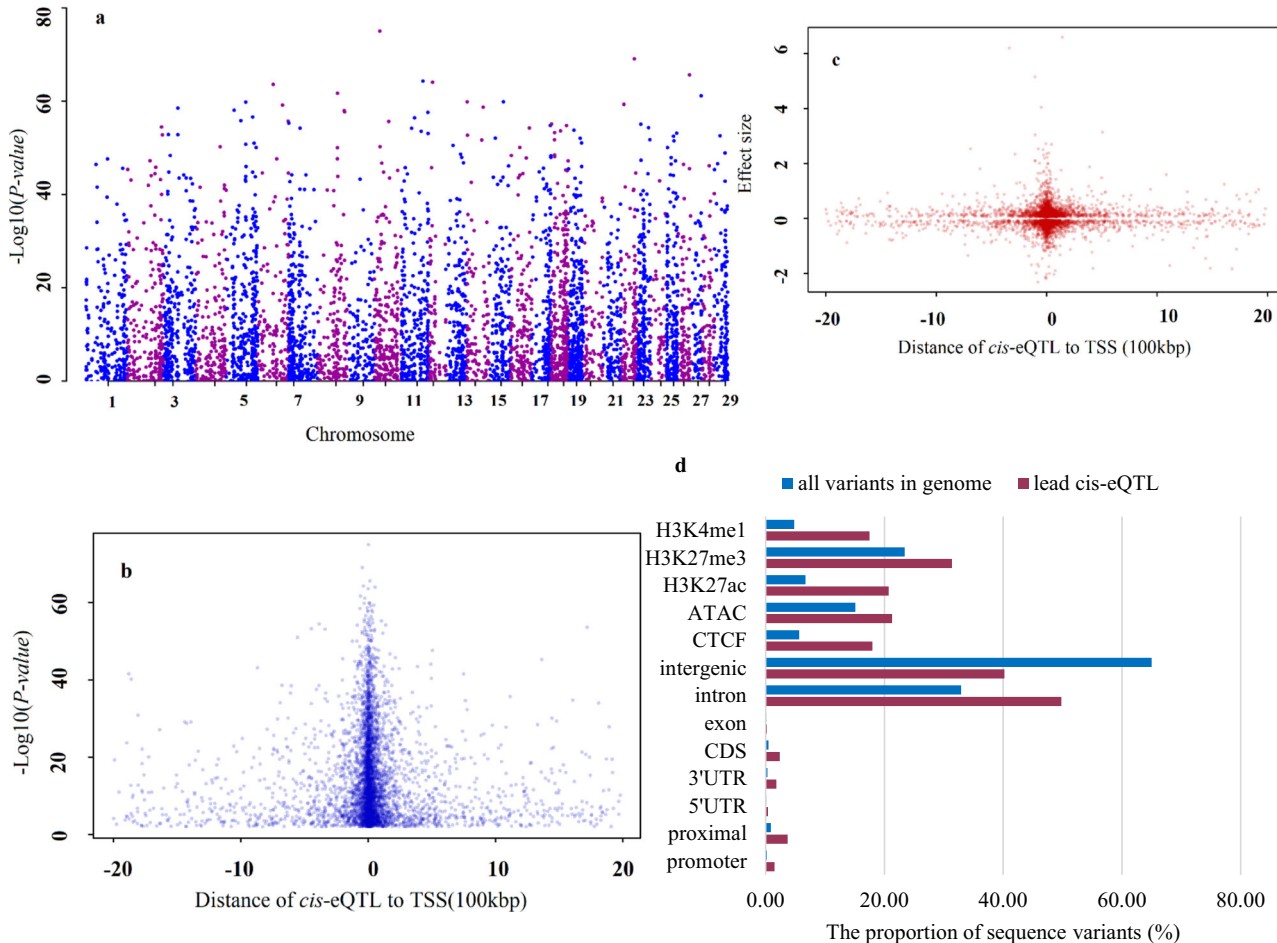

**Fig. 5 | Results of the *cis*-eQTL analysis. a** Manhattan plot of lead cis-eQTLs for eGenes (FDR < 0.01), **b** The relationship between lead cis-eQTLs adjusted *P*-value and distance to transcription start site (TSS), **c** The relationship between lead cis-eQTLs effect sizes and distance transcription start site (TSS), **d** Annotations of the lead cis-eQTLs compared to the proportion of all sequence variants.

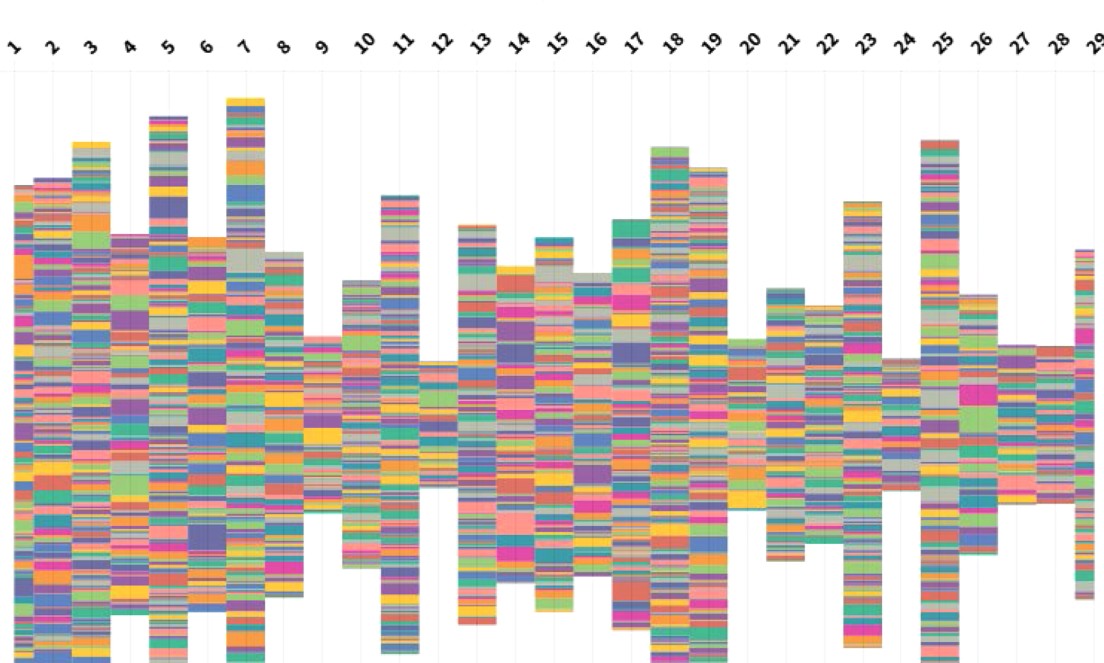

**Fig. 6 | Distribution of cis-eGenes (genes with significant cis-eQTL) and number of cis-eQTLs discovered for each cis-eGene in whole blood across different chromosomes.** Each coloured box represents a different cis-eGene and the depth of coloured box represents the number of identified cis-eQTLs at global-FDR threshold 0.01.

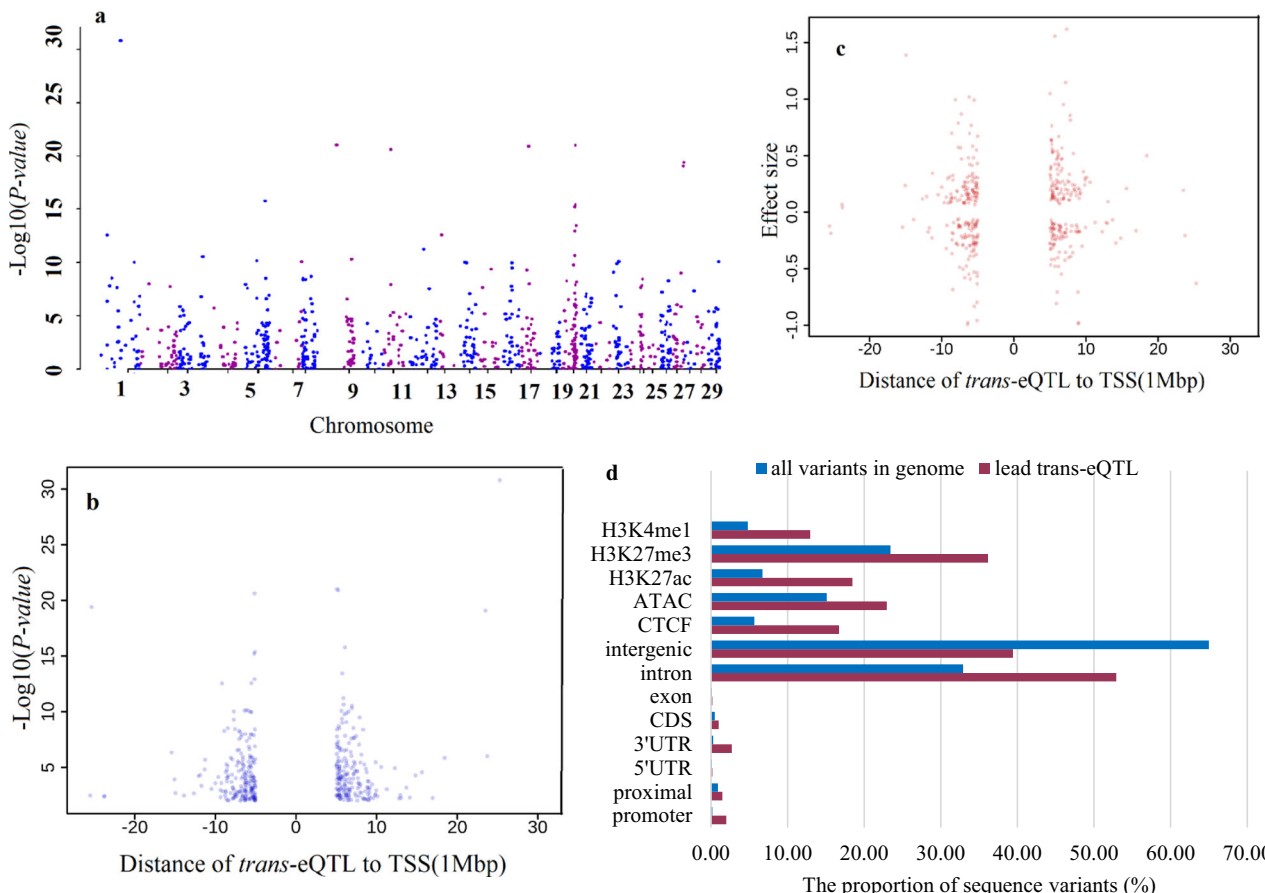

**Fig. 7 | Results of the *trans*-eQTL analysis. a** Manhattan plot of lead *trans*-eQTL for eGenes (FDR < 0.01), **b** The relationship between lead *trans*-eQTLs adjusted *P*-value and distance to transcription start site (TSS), **c** The relationship between lead *trans*-eQTLs effect sizes and distance to transcription start site (TSS), **d** Annotations of the lead *trans*-eQTLs compared to the proportion of all sequence variants.

addition, the *HELB* gene is involved in DNA damage response including exposure to ultra-violet light, and thus, is relevant for tropical adaptation[41].

Several reasons may have contributed to a low level of overlap between lead GWAS SNPs and eQTL detected in our study, including a lack of power in the GWAS in the discovery population as a result of a sample size of 2119 cattle. Another reason for the limited overlap between eQTL and GWAS peaks may be due to the fact that some of the genes affecting fertility are not expressed in the whole blood of heifers or cows. Although it is difficult to sample other tissues at scale in the same way as blood, future eQTL studies in foetal tissues are recommended.

## Pathway analysis

Next, we conducted a follow-up study of the final combined list of 87 significant genes associated with fertility in cattle (Methods; Supplementary Data 10). The most strongly enriched pathway identified by Gene Ontology analysis of genes associated with fertility was *Extracellular Matrix (ECM)-receptor interaction pathways* ($P$-value $< 6.2 \times 10^{-4}$, FDR $< 0.05$) and 5 genes were identified including *CD44, ITGA6, ITGB5, LAMA2,* and *SV2B*. In agreement with a previous study, this outcome has highlighted that the balance of ECM degradation and remodelling is vital to the regulation of maternal–foetal interface including menstrual cycling, decidualization, embryo implantation, and pregnancy maintenance and disorders in these events may eventually lead to pregnancy failures[43].

## Common genes regulate fertility in cattle and humans

We next investigated the overlap of genes affecting fertility in cattle and humans using the same gene list used for Gene Ontology analysis (Methods; Supplementary Data 10). Of the 87 cattle fertility-related genes investigated, eight genes (*BMP4, HELP, KCNIP1, KRT222, L3MBTL3, LAMA2, RBBP8, SV2B*), and four genes (*SMC1B, THSD7B, SATB2,* and *LRP1B*) were previously reported to be associated with age at natural menopuase[1] and age at menarche[18] in humans, respectively, which is higher than expected by chance (Fisher's exact test; $P$-value $< 0.05$; Supplementary Data 11, 12, 13). Of the 8 genes associated with fertility in cattle and age at natural menopause in humans, 4 genes (*KCNIP1, LAMA2, KRT222,* and *SV2B*) were reported as being located at the centre of a multi-tissue co-expression network interacting with many other genes near the age at natural menopause associated variants in humans[1].

## Conclusion

In cattle, poor fertility is one of the most common reasons for culling, and poor conception rates can affect the profitability and sustainability of both beef and dairy production. Our results provide an extensive resource of *cis*-eQTLs and *trans*-eQTLs at the gene level which may be useful for elucidating the biological underpinnings of many SNPs associated with fertility traits, as we have demonstrated for several genes. The mutations we have identified, particularly those significant in the validation tests, should be included on commercial SNP arrays to improve accuracy of genomic estimated breeding values for fertility. We have also highlighted that gene expression in blood may only help to identify some of the mutations affecting fertility. Future eQTL studies in foetal tissues is recommended, as it appears several genes responsible for allocating foetal resources to growth or reproduction are - expressed at much higher levels at the foetal stage. We found significant overlap among the genes close to the SNP that were significant in our SMR and GWAS analysis, and genes implicated in human GWAS for age at natural menopause and age at menarche. Furthermore, we highlight that the pathway associated with the extracellular matrix (ECM) degradation, critical for the regulation of the maternal–foetal interface including menstrual cycling, embryo implantation, and pregnancy maintenance, was enriched for significant cattle fertility-related genes including *CD44, ITGA6, ITGB5, LAMA2,* and *SV2B*.

## Methods

### Phenotypic data

For this study, females born between 1998 and 2018 in a Central Queensland Brahman cattle herd were assessed (discovery population; Supplementary Data 1). Heifers (1st parity) and cows (2+ parity) were part of a seedstock herd that was maintained on tropical pastures at latitude 20–22⁰ South. There was a singular emphasis on fertility where failure to give birth to a calf was the primary culling criterion. Four fertility-related traits were recorded. Heifer pregnancy status (preg_st) was recorded as a binary trait (1 = successful, 0 = unsuccessful) indicating whether a heifer conceived before three years of age. Foetal age in weeks (wks_preg) was recorded via manual palpation at pregnancy diagnosis, to assess the age of the foetus, for all heifers born in 2011 and later. This trait is a measure of fertility, as those heifers with older foetuses have likely cycled and conceived soon after the bull entered the paddock, whereas those with younger foetal ages have cycled and conceived later. Age at first calving (AFC) was only available for heifers with a recorded birth date and was calculated as the difference in days between first calving and birthdate. Days to calving (DTC) is a routinely recorded trait in Australian Brahmans and was defined as the number of days between the date of first bull exposure at the beginning of the breeding season and calving date.

### Genotyping

Heifers and cows were genotyped with the BovineSNP50 BeadChip (Illumina, San Diego, CA) or the Neogen TropBeef 50k array. A detailed description of the genotype quality control can be found in a previous study[44]. Genotypes were imputed up to 709,768 SNPs (Bovine HD array) using findhap4[45] and a reference panel of 4650 cattle genotyped for the Bovine HD array (Illumina, San Diego, CA) from relevant breeds, including 150 Brahman cattle. All genotypes were then imputed to WGS variants using the 1000 Bull Genomes Run8, TaurIndicus reference [46], with 600 Holstein and 400 Simmental animals removed to avoid over-representation of these genomes in the imputation, such that 1261 whole genome sequenced animals remained. Eagle[47] was used for phasing and Minimac3[48] for imputation. Sequence variants with fewer than 4 copies of the minor allele were removed prior to imputation in an attempt to avoid including sequencing errors in the data set. After this filter, 49,779,229 variants were imputed. The imputation accuracy of Brahman genotypes using 1000 Bull Genomes Run8 was greater than 0.85 for variants with minor allele frequency (MAF) > 0.05 (Supplementary Fig. 2; Supplementary Data 14). Only 29 autosomes and variants with imputation accuracy of greater than 0.4[49] and MAF > 0.01 (31,140,417) were used for the GWAS study.

### Genome-wide association analysis

For the discovery population with the four fertility traits, a linear mixed model was performed using the fastGWA approach in GCTA software[50], fitting each sequence variant as a covariate, one at a time, and testing for association with each trait as follows:

$$\boldsymbol{y} = \boldsymbol{1}_n \mu + \boldsymbol{X}\beta + \boldsymbol{Z}g + \boldsymbol{W}_i \alpha_i + \boldsymbol{e},$$

where $\mathbf{y}$ is the vector of phenotypic values of the animals (up to 2,119 animals; Supplementary Data 1), $\boldsymbol{I}_n$ is an n × 1 vector of 1s (n=number of animals with phenotypes), μ is the overall mean, $\mathbf{X}$ is an n × x matrix of fixed covariates, β is a length x vector of fixed effects, $\mathbf{Z}$ is a design matrix for the random additive genetic effects, and g is a vector of random additive genetic effects assumed to be distributed as $\sim N(0, \mathbf{G}\sigma_g^2)$, where $\mathbf{G}$ is the genomic relationship matrix (GRM) calculated from high-density genotypes using the GCTA software. $\boldsymbol{W}_i$ is a vector of genotypes for each animal at the $i^{th}$ variant, $\alpha_i$ is the coresponding additive effect (fixed effect) of the variant. The genotypes at each locus were coded as 0, 1, or 2, representing the number of copies of a particular allele carried by an individual. $\mathbf{e}$ is a random vector of length n as $\sim N (0, \sigma_e^2 \mathbf{I})$, where $\sigma_e^2$ represents non-genetic variance due to non-genetic effects assumed to be acting independently on animals. The

choice of fixed covariate effects for continuous and binary traits was done using the *lm* and *glm* functions in R, respectively. For all four traits, year of birth and contemporary group were considered as fixed covariates. The contemporary group was defined as the year-season effects, with 7 levels. Moreover, for trait AFC, calving success defined as 0 and 1, based upon whether they calved before or after 900 days of age was considered as a fixed effect. Also, for trait DTC, the effect of heifer age of joining (days) was fitted as continuous covariate fixed effect. The GRM was generated using variants with MAF higher than 0.01 (609,878 SNPs) using the Bovine HD data.

We performed a conditional multi-trait analysis (CMT-GWAS) according to a previously described approach[8] using WGS variant effects estimated from the four single-trait GWAS to identify pleiotropic variants that affected fertility traits. The multi-trait analysis (M-GWAS) $X^2$ statistic with 4 degrees of freedom (equal to the number of traits analysed) was calculated as: $X^2 = \mathbf{t_i V^{-1} t_i}$, where $\mathbf{t_i}$ is a vector of the signed *t*-values of the effects of the $i^{th}$ sequence variants for the 4 traits and $\mathbf{V^{-1}}$ is the inverse of the $4 \times 4$ correlation matrix where the correlation was calculated over all estimated sequence variant effects (signed *t*-values) between each pair of traits. The CMT-GWAS approach cycles back and forward between the single-trait GWAS for all traits and M-GWAS to re-test variants conditional on jointly fitting the most significant putative causal variants from independent QTL (where we defined significant as $P < 5 \times 10^{-6}$). To determine independent sequence variants, first, the most significant M-GWAS variant from each chromosome was selected and added to the list of putative causal variants. If the pairwise LD between this variant and any other significant variant on the same chromosome was greater than 0.1, these other variants are considered as potentially tagging the same causal variant and were not considered as independent QTL for this cycle. Then, from the remaining significant variants in LD, $r^2$ less than 0.1, the next most significant variant was selected on each chromosome, LD was tested between these and the remaining significant variants, and so on, until no more significant variants were identified in this cycle.

FDR was calculated as $\frac{P(1-\frac{A}{T})}{(\frac{A}{T})(1-P)}$, where $P$ is the p-value tested, $A$, is the number of SNP that were significant and the p-value tested and $T$ is the total number of SNP tested[12].

## Validation

The validation population consisted of 27,707 indicine, Bos taurus, and crossbred heifers genotyped with the Neogen TropBeef 50k array with records for heifer puberty[44]. Heifer puberty was defined as a binary trait: 0 indicated the absence of a corpus luteum at approximately 600 days, while 1 indicated the presence of a corpus luteum at approximately 600 days[51]. All animals were imputed to WGS as described for the discovery population.

To determine the proportion of variation explained by the 225 potential causal variants identified in single/multi-trait GWAS of the discovery population, genotypes for the 225 SNPs were extracted for the indicine animals in the validation population (animals with indicine content equal greater than 0.80) and a GRM was estimated using these SNPs. The proportion of variance explained by this matrix was determined for the heifer puberty trait by fitting the model;

$$y = \mathbf{1}_n \mu + X\beta + Zu + e$$

where $y$ is a vector of phenotype, $\mathbf{1}_n$ is a vector of ones, $\mu$ is the mean, $\mathbf{X}$ is an n × x matrix of fixed covariates, $\beta$ is a length x vector of fixed effects, $\mathbf{Z}$ is a design matrix allocating phenotypes to animals, $u$ is a vector of breeding values and $e$ is a vector of random residuals. The effect of contemporary group covariates and indicine content were fitted as class and continuous covariate fixed effect, respectively. The breeding values $u$ were assumed to be derived from a normal distribution $u \sim N(0, G\sigma_g^2)$, where $G$ is the GRM created from genotypes at the 225 potential causal variants and $\sigma_g^2$ is the additive phenotypic variance explained by the 225 potential causal variants. Variance components were estimated with *reml* function in GCTA[50]. To determine the proportion of variance expected to be explained by chance,

another 225 variants were randomly sampled from the sequence data, and the model above was fitted. This was done five times, and the proportions of explained variance were averaged.

As an additional analysis, the significance of 225 potential causal variants identified from single/multi-trait GWAS in the discovery population was assessed in genome-wide summary statistic results for heifer puberty in the validation population[52].

## *cis-eQTL* mapping

Gene expression levels in whole blood samples were profiled by RNA-seq. Blood samples were collected from 489 indicine heifers and cows in the discovery population in accordance with ethics approved by the University of Queensland Animal ethics committee (SAFS /262/20, SAFS/253/20, QAAFI/269/17, QAAFI/270/17). Pre-pubertal heifers born in 2018 ($n = 116$) selected for RNA sequencing were stratified by management cohort, date of birth, and sire. All two-year-old heifers born in 2016 ($n = 301$) from this herd were sequenced. A small proportion of these heifers were pregnant. The mature cows ($n = 72$, number of calves ranging from one to five) in this study were born between 1999 and 2009, and were selected for RNA sequencing if they were genotyped, had yearly production records, and were balanced across sire. All mature cows and two-year-old heifers had been exposed to bulls in a natural service, multi-sire breed season lasting for five months. At the time of sample collection, the lactation and pregnancy status (recorded as foetal age in weeks) of each female was determined. Samples were collected over three separate days.

Blood was collected from the tail vein in 10 ML EDTA vacutainers. A 500 ul aliquot of whole blood was immediately drawn from the vacutainer and added the a Qiagen RNAprotect Animal Blood Tube. The tubes were then incubated at ambient temperature for 2 h as per manufacturers instructions. After incubation, the tubes were transported at −20 ℃, then stored at −80 ℃ until extraction. RNeasy Protect Blood Kit (QIAGEN) was used to extract total RNA from whole blood samples. RNA purity and quantity were evaluated with a Nanodrop ND-1000 spectrophotometer (v.3.5.2, Thermo Fisher Scientific) and QubitTM 4.0 Fluorometer with the Qubit RNA BR (broad-range) assay kit (Thermo Fisher Scientific). The assessment of RNA integrity was performed using the LabChip GX assays (Perkin Elmer). RNA with integrity number greater than 6.9 was used for library preparation for sequencing.

All RNA samples were sent to the Ramaciotti Centre for Genomics (UNSW Sydney, Australia) for library preparation and sequencing using TruSeq Stranded mRNA. Stranded paired-end RNA-seq libraries were sequenced on a 2 × 100 bp paired-end NovaSeq6000 run with an S4 flowcell.

We used the pipeline described in Chamberlain et al.[53] to process the gene expression data. Briefly, QuadTrim[54] was used to trim and filter poor-quality bases and sequence reads. Adaptor sequences were trimmed based on pair overlap and bases with a quality score of < 20 were removed from the ends. Reads with a mean quality score less than 20, greater than 3 N, greater than three consecutive bases with a mean quality score of less than 15, or a final length of fewer than 50 bases were discarded. After quality control, the mean depth was 46,618,531 reads, with a minimum and maximum of 15,424,975 and 68,193,182, respectively. To avoid mapping bias, ARS-UCD1.2 bovine genome[55] assembly was masked at all known variant sites from 1000 bull genomes run 8 with an allele not present at that position. High-quality raw reads were aligned to this masked reference genome with STAR[56] using the 2-pass method. The gene counts were extracted with FeatureCount[57]. Genes with expression counts less than 3 counts per million mapped (CPM) in at least 25% of the population were excluded, leaving 10,455 genes included in this analysis.

Imputed WGS data for these animals was filtered based on MAF greater than 0.05 and imputation accuracy > 0.4 for the cows used in the analysis (no LD pruning was performed) resulting in 24,902,617 variants for eQTL analysis. For the *cis*-eQTL analysis, the association between gene expression and each variant within a 2 Mb upstream and downstream window around the gene start site was estimated using a mixed linear model in OSCA[34], fitting the genotype of the variant, birth year-contemporary

group, lactation status, weeks pregnant and the first four principal components of gene expression as fixed effects, to control for batch effects. The animal was fitted as a random effect, $\sim N(0, \mathbf{G}\sigma_g^2)$, where $\mathbf{G}$ was the genomic relationship among the animals derived from Bovine HD genotypes to account for the population structure.

## Multiple-testing correction for *cis*-eQTL mapping

A two-step procedure was employed to perform hierarchical multiple testing correction[35]. First, p-values of all *cis*-SNPs were adjusted for each gene separately based on Bonferroni method for multiple SNPs tested (locally adjusted p-value; Step 1). The locally adjusted p-value was calculated by multiplying each p-value to the number of SNPs in the corresponding *cis* window for the tested gene. Next, the eGenes were identified as those having the minimum locally adjusted p-value from Step 1, below the p-value threshold of 0.01, and adjusted for the number of genes tested (10,455). The *cis*-eQTLs were identified for each eGene as SNPs with a locally adjusted p-value from Step 1 lower than the p-value threshold of 0.01 adjusted for the number of genes tested (10,455). Supplementary Fig. 3 shows the enrichment of top 20 most significant Gene Ontology terms among the genes tested.

## *cis-eQTL* inter-validation

To assess the reproducibility of eQTL findings, the animal cohort was randomly divided into two distinct subsamples, comprising 244 and 245 animals each, respectively. The same statistical model, as previously outlined for *cis*-eQTL mapping was implemented independently in each subsample. Pearson correlation coefficients were then computed to evaluate the concordance of lead eQTL effects between subsamples and between subsample and original sample across all eGenes. Additionally, the frequency of shared *cis*-eGenes between subsamples and the original sample was quantified to measure consistency across subsamples and subsample and the original dataset.

## trans-eQTL mapping

We aimed to perform *trans*-eQTL mapping for significant variants identified through *cis*-eQTL or GWAS analysis. We specifically targeted genes where the variant was located more than 5 Mb away from the gene on the same chromosome. So, the most significant *cis*-eQTLs for each eGene ($n = 4376$, FDR < 0.01) were selected. Additionally, we added 225 genome-wide significant GWAS variants from the single trait and CMT-GWAS model described above, applying a significance threshold $P \le 5 \times 10^{-8}$. We also added variants within 100 bp on either side of each selected variant, which yielded 14,268 variants. We tested the associations between all selected variants and genes that were at least 5 Mb away to ensure that they did not tag a *cis*-eQTL effect on the same chromosome. *Trans*-eQTL mapping was performed using the same model described for *cis-eQTL* mapping. A two-step procedure based on the Bonferroni method and FDR threshold of 0.01 described above was employed to perform hierarchical multiple testing correction for the *trans-eQTL* mapping.

## Integration of eQTLs with regulatory regions

The lead *cis*- and *trans*-eQTLs overlapped with regulatory regions identified from different ChIP-seq and ATAC-seq datasets from a range of tissues. ATAC-seq and ChIP-seq peaks for H3K27ac from bovine liver were downloaded (ArrayExpress accession number E-MTAB-2633)[16]; H3K27ac, H3K4me1, H3K27me3, and CTCF from a range of bovine tissues including liver, lung, spleen, skeletal muscle, subcutaneous adipose, cerebellum, brain cortex, and hypothalamus (GSE158430)[17].

## Integration of GWAS results with eQTL

We applied the Summary data–based Mendelian Randomization analysis (SMR) method[40] to test for causation of mutations associated with the four fertility traits and the expression level of each gene across the whole genome using summary data from the single-trait GWAS in the discovery population and *cis*-eQTL studies. We selected the top associated eQTL at $P < 5 \times 10^{-8}$ and QTLs at $P < 5 \times 10^{-3}$ as an instrument for an SMR analysis.

## Follow-up study of fertility related genes in cattle

First, we performed Gene Ontology analysis on the list of genes associated with fertility using the DAVID web server[58] and considered the entire taurine gene set as a reference data set. The genes associated with fertility included: (1) genes that showed associations with fertility traits ($P_{SMR} < 3.5 \times 10^{-4}$), (2) genes overlapped or within ±5 kb of the most significant GWAS variants for all fertility traits ($P < 5 \times 10^{-8}$), and 3) genes nearest the top four most significant variants from GWAS analysis of the validation trait (heifer puberty) (Supplementary Data 10).

Then, we estimated the number of orthologous genes in or close to GWAS peaks for human fertility that were also located in or close to GWAS peaks for fertility in cattle, and tested if the overlap was greater than would be expected by chance. Two human fertility traits, age at natural menopuase and menarche, in about 201,323 women of European ancestry (data obtained from Supplementary Tables 2, 17, and 18[1]) and 368,888 women (data obtained from Supplementary Tables 2[18]) were considered, respectively. The Fisher's exact test was used for the analysis of significance level (Supplementary Data 11, 12).

## Statistics and reproducibility

We performed single-trait GWAS analysis and conditional multi-trait GWAS analysis in up to 2119 cows and heifers for four fertility traits of DTC, AFC, preg_st, and wks_preg (discovery population; indicine subspecies). We used a validation population consisting of 27,707 indicine, taurine and crossbred heifers genotyped with the Neogen TropBeef 50k array, imputed to whole genome sequence as described for the discovery population, with records for heifer puberty, which was the presence or absence of a corpus luteum at approximately 600 days assessed by ultrasound scanning. The eQTL analysis was performed on whole blood in 489 heifers and cows from the discovery population. We assessed the reproducibility of *cis*-eQTL analysis by splitting the data into random subsamples of 244 and 245 animals.

## Reporting summary

Further information on research design is available in the Nature Portfolio Reporting Summary linked to this article.

## Data availability

Supporting findings are included in the published article (including Supplementary Data 1–14 and Supplementary Figure). The fastq files for gene expression profile were deposited in ENA with the project accession PRJNA1090634. The summary statistics for GWAS analysis are available at figshare[59].

## Code availability

Script developed to perform multi-trait GWAS analysis is available in GitHub[60].

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

## Acknowledgements
We acknowledge financial contributions from ARC (LP16101626) and NG (P.PSH.0833) Projects. We are also thankful to Alf Collins Snr (Collin's Belah Valley, Marlborough, Central Queensland) who provided animals and historical fertility records used in the current study. We would like to acknowledge the 1000 Bull Genomes project for collecting and building a resource of sequenced key ancestor bulls for the bovine research community which was used as the reference dataset for imputation of genotypes in this study.

## Author contributions
M.F., B.J.H., and M.E.G. conceived and designed the study. B.J.H. and E.M.R. collected the blood samples. B.N.E. collected and curated all phenotypic data and genotype data from the gene expression/discovery population. MF performed the GWAS, eQTL, SMR analysis, and enrichment analysis. E.A.K., and L.T.N. performed RNA sample extraction. E.M.R., B.N.E., and L.T.N. performed the RNAseq quality control and alignment. A.J.C. performed RNAseq gene counting. M.F. wrote the paper. ACSnr provided cattle for the experiment and assisted in the experimental design. G.F. and S.S. helped design the experiments and collect data. MD provided input on gene function and reproductive biology. All authors read and approved the final manuscript.

## Competing interests
Alf Collins Snr has provided partial funding for the ARC project. Other authors declare that they have no competing interests.

## Ethical approval
We have complied with all relevant ethical regulations for animal use.

## Additional information

**Peer review information** *Communications Biology* thanks Genome-wide association and expression of quantitative trait loci in cattle reveals common genes regulating mammalian fertility.

