## [Peer Review File · Communications Biology]

Reviewers' comments:

Reviewer #1 (Remarks to the Author):

This study links results from GWAS for fertility traits with expression QTL analysis from whole blood to identify key genes affecting fertility in cattle. Two populations of cattle are used for the analysis a 'discovery' population of *Bos indicus* cattle and a 'validation' population including *Bos taurus* and *Bos indicus* animals. The authors identify genomic regions, eQTLs and genes associated with fertility and perform a comparative analysis of these with genes known to influence fertility in humans.

The results of the study are interesting and the SNPs and eQTLs are a useful resource for the community and cattle breeding more broadly. The narrative is very difficult to follow however as details are missing throughout and the datasets are described too late in the manuscript to really follow how and why they are used. More clarity is required in the rationale for using each dataset and its relevance and this will make the narrative much easier to follow.

Abstract

In the abstract could the authors please mention the breed of cattle or indicate whether the study relates to dairy or beef cattle.

Also in the abstract could the authors explain the trait 'foetus age in weeks' is this the time in weeks that the heifer carried the foetus e.g. the duration of the pregnancy?

Introduction

Line 52-53 Some more explanation is required here to indicate that SNP chips, with specific pre-defined content, are less likely to include variants with low heritability associated with fertility.

Could the authors mention the breed or sub-species in the introduction and describe the study population further? Were all the datasets from the same population including the 28K heifers the results were validated in? This information is provided in the results but this paragraph in the introduction requires some additional detail added to make the rationale for the study clearer. It would even be useful to define the discovery and validation populations here I think.

In the introduction it is also important to mention the CHIP and ATAC data and the analysis of the eQTLs in relation to linking fertility traits to functional genomic regions.

Results and Discussion

I realise it is a prerequisite of the journal but having the methods after the results and discussion makes the narrative for this paper quite difficult to follow as there are several different datasets and populations being analysed. This is why I think describing the study populations at the end of the

introduction might make following the narrative easier.

Line 112-113 Please add details of the species when describing expression levels of genes here and throughout the manuscript. The same is true for the GWAS results where often the species is not reported and this makes the manuscript really hard to follow.

Line 128 to 130 Make sure when it's the results of this study that this is referred to so the narrative is easier to follow. Without referring to the gene expression results relating to the cohort of cattle with expression profiles from whole blood analysed in this study it is really difficult to follow which study is being referred to.

Line 134 The expression of PLAG1 is only measured in whole blood in this study though, could it not be higher in other tissues, such as liver that are associated with growth, that are not measured and in fetal tissues as shown in fig 4 and Fang et al. 2020?

Line 156-157 Could the authors determine if this is the case by performing the analysis only with *Bos indicus* individuals in the validation population?

Line 165-166 Include the breed of the heifers here or if they were *Bos indicus* so the narrative is easier to follow.

Line 165- 166 More detail in Fig 1 on the populations would really help here to show which dataset and breeds etc were used for which component of the analysis. Adding that the cattle in the discovery population were *Bos indicus* would also be useful to make clear here.

line 207-207 More description is required here. What is the functional relevance of the lead trans eQTLs being located in ATAC-Seq or CHIP peaks? What is the significance of the datasets used e.g. for the purposes of annotation of the bovine genome as a resource to link complex traits to functional genomic regions.

For figure 3 could any of the GWAS hits be annotated?

Methods

Including some of this information about the study population earlier in the manuscript would make the rationale for the study and narrative of the results and discussion much clearer.

Could more information in Fig 1 be included showing the different breeds in the datasets used for each part of the analysis?

Were the red blood cells lysed before the RNA extractions were performed? Would the expression profiles be predominantly associated with white blood cells?

Line 411 After filtering for expression 'of the 10,455 genes included in this analysis' were these

genes particularly associated with any functional roles e.g. immunity?

The datasets haven't been deposited in the public repositories yet but the authors indicate where the accession will be added in the manuscript. This should ideally be done before the manuscript is resubmitted.

Specific line changes

Line 50 change 'demonstrates' to 'demonstrating'

Line 82 Add 'in humans' after 'menarche'

Line 160-161 Change to 'The current results provide'

Line 183-184 Include whether this study is on cattle.

Line 184-185 and presumably also sex and age of animals?

Line 190-192 Provide details of the genome and annotation used to define these regions.

Line 193-194 Add citation or accession and some description of the ChIP data, which tissue etc?

Line 206 Add 'a' before 'previous' and add whether the study is on cattle.

Line 218 Delete 'variants'?

Line 220 On the expression data from whole blood?

Line 230 Add reference for the previous study.

Line 237 Change 'the' to 'a'

Line 241 Add that the effect might be more obvious in other tissues, but is difficult to sample other tissues at scale in the same way as blood.

Line 275 Remove the extra 'the'

Line 290 I can't see how fetal age in weeks is a heritable trait as it is explained here?

Line 392-393 How were the blood samples stored and preserved prior to RNA extraction?

Line 400 What was the sequencing depth per sample?

Line 452 Please include the ENA accessions for these datasets and also add the cattle gene expression atlas reference here for the data shown in Fig 4.

Reviewer #2 (Remarks to the Author):

Forutan et al. studied the genetic underpinning of female fertility, an economically important yet challenging trait in cattle. They use multiple proxy phenotypes in two cattle populations and integrate RNA seq data to identify 87 genes putatively associated with female fertility. They further show evidence for a shared pool of genes regulating fertility in cattle and human. The paper is thus of interest to the broad readership of the journal.

The paper is interesting and generally well written. However, I have the following concerns

1. Authors motivate integration eQTL data (Line 217) by stating significant GWAS loci which are also eQTL are more likely to be causal variants. Is there a reason why splicing QTL were not studied?

Given there is limited overlap between GWAS and eQTL signals (Line 237), it might be worth looking into splicing QTL.

2. It is not clear in the result section that the trans-eQTL scan was limited to variants significant in GWA and cis-eQTL mapping analyses. A discussion on how many cis-eQTL showed trans effect is perhaps interesting.

3. The proportion of variance explained by the 255 associated variants is reported only for one trait (heifer puberty). A discussion on variance explained for the other four fertility traits could be interesting.

Few other minor points

1. In methods, (example #358) heifer puberty is referred to as CL600.

2. Line #80: “some overlap in” – perhaps authors could provide numbers here.

3. Line #124: Not clear what do authors mean by “full GWAS”.

Reviewer #3 (Remarks to the Author):

The manuscript reports identification of genes affecting fertility in cattle. Fertility traits are economically important to beef cattle production. The authors conducted a comprehensive study and the results are very valuable to both the science community and the industry. In general, the experimental design and data analyses were well described and the results were well presented and discussed. The manuscript can be accepted for publication after addressing the following comments:

Line 154, “these loci explained 2.3% of the variation”. Was it phenotypic or genetic variance? Please clarify in the manuscript.

Line 187. “Seventy-eight percent of the lead cis-eQTLs were close to the respective gene start site”. How many cis-eQTL were within the respective gene? i.e. a DNA variant within a gene was found to be associated with the gene expression of the same gene.

Line 311 “Only 29 autosomes and variants with imputation accuracy of greater than 0.4 and $MAF > 0.01$ (31,140,417) were used for GWAS study”. An imputation accuracy of 0.4 seems very low (also in line 413). Please justify why an accuracy of greater than 0.4 was used. In addition, please briefly describe how the imputation accuracy of DNA variants was assessed.

Line 324, please indicate how the genotypes were coded and what variant effect (i.e allele substitution or additive effect) was estimated? Based on supplementary tables, it looks like that allele substitution effect (b) was estimated. But please include the SNP effect estimation information in the manuscript.

Line 327, Please describe how the binary trait “Heifer pregnancy status” was analyzed.

Line 328, please describe how was the contemporary groups defined, and how many levels of the contemporary groups.

Line 330, “for trait AFC, calving success defined as 0 and 1, based upon whether they calved before or after 900 days of age was considered as a fixed effect”. Does this mean if the cow calved before 900 days of age, its calving success was assigned “1”, or otherwise as “0”. Please justify why calving success was considered as a fixed effect for AFC.

Line 366, please describe phenotype values of the validation population in more detail. Were they pre-adjusted for non-genetic effects?

Line 380. Did the 489 Bos indicus heifers and cows used for expression analysis have the same or similar four fertility phenotypes? If yes, did the 489 cattle have similar mean and ranges of variation as the population used for the GWAS?

Table S1, it would be more informative to add mean, range of phenotypic values, SD of the four traits and their estimates of genomic Heritability from the GWAS?

Table S2 and Table S4, It would be more informative to add SNP annotation (A1, A2) and indicate which SNP allele is the minor allele with the allele frequency, i.e. similar to Table S6 on SNP information (A1 A2).

Reviewers' comments:

Reviewer #1 (Remarks to the Author):

This study links results from GWAS for fertility traits with expression QTL analysis from whole blood to identify key genes affecting fertility in cattle. Two populations of cattle are used for the analysis a 'discovery' population of *Bos indicus* cattle and a 'validation' population including *Bos taurus* and *Bos indicus* animals. The authors identify genomic regions, eQTLs and genes associated with fertility and perform a comparative analysis of these with genes known to influence fertility in humans.

The results of the study are interesting and the SNPs and eQTLs are a useful resource for the community and cattle breeding more broadly. The narrative is very difficult to follow however as details are missing throughout and the datasets are described too late in the manuscript to really follow how and why they are used. More clarity is required in the rationale for using each dataset and its relevance and this will make the narrative much easier to follow.

authors: Thanks for your valuable comments. More details have been added throughout to make the narrative clearer.

Abstract

In the abstract could the authors please **mention the breed of cattle or indicate whether the study relates to dairy or beef cattle**. Also in the abstract could the authors explain the trait '**foetus age in weeks**' is this the time in weeks that the heifer carried the foetus e.g. the duration of the pregnancy?

authors: The breed of cattle used for the analysis and trait 'foetus age in weeks' have now been described in the abstract.

Introduction

Line 52-53 Some more explanation is required here to indicate that SNP chips, with specific pre-defined content, are less likely to include variants with low heritability associated with fertility.

authors: More explanation has been added to this paragraph.

Could the authors mention the breed or sub-species in the introduction and describe the study population further? Were all the datasets from the same population including the 28K heifers the results were validated in? **This information is provided in the results but this paragraph in the introduction requires some additional detail added to make the rationale for the study clearer**. It would even be useful to define the discovery and validation populations here I think.

authors: The discovery and validation populations have now been more clearly described in the introduction.

In the introduction it is also important **to mention the ChIP and ATAC data and the analysis of the eQTLs in relation to linking** fertility traits to functional genomic regions.

authors: Additional detail has been added in the introduction to describe this.

Results and Discussion

I realise it is a prerequisite of the journal but having the methods after the results and discussion makes the narrative for this paper quite difficult to follow as there are several different datasets and populations being analysed. This is why I think describing the study populations at the end of the introduction might make following the narrative easier.

authors: The study population was described at the end of the introduction to make the narrative easier to follow.

Line 112-113 Please add details of the species when describing expression levels of genes here and throughout the manuscript. The same is true for the GWAS results where often the species is not reported and this makes the manuscript really hard to follow.

authors: Details of the species was added throughout the manuscript.

Line 128 to 130 Make sure when it's the results of this study that this is referred to so the narrative is easier to follow. Without referring to the gene expression results relating to the cohort of cattle with expression profiles from whole blood analysed in this study it is really difficult to follow which study is being referred to.

authors: More detail has been added on the expression profile study in multiple places to hopefully make the manuscript easier to follow

Line 134 The expression of PLAG1 is only measured in whole blood in this study though, could it not be higher in other tissues, such as liver that are associated with growth, that are not measured and in fetal tissues as shown in fig 4 and Fang et al. 2020?

authors: While the gene expression was solely assessed in whole blood in the present study, our previous research (Forutan M, et al. 2021, "Evolution of tissue and developmental specificity of transcription start sites in *Bos taurus indicus*") uncovered that the expression of PLAG1 is diminished in adult liver compared to fetal liver, suggesting challenges in identifying eQTL in mature animals. This has been added to the text.

Line 156-157 Could the authors determine if this is the case by performing the analysis only with *Bos indicus* individuals in the validation population?

authors: Thank you for your valuable input. We ran the prediction model using only cattle with high *Bos indicus* content (≥ 0.80) in the validation population. We did observe a small increase in the amount of variation accounted for by the 225 potential causal variants in the population with a high content of *Bos indicus* compared to the entire validation population, which includes *Bos indicus*, *Bos taurus*, and crosses. This section has been updated to include the new results.

Line 165-166 Include the breed of the heifers here or if they were *Bos indicus* so the narrative is easier to follow.

authors: The breed of cattle used for the analysis is now described in heading.

Line 165- 166 More detail in Fig 1 on the populations would really help here to show which dataset and breeds etc were used for which component of the analysis. Adding that the cattle in the discovery population were *Bos indicus* would also be useful to make clear here.

authors: The heading was changed to "cis-eQTLs and trans-eQTLs in *Bos indicus* cattle" and the caption for Fig 1 was edited to "Flow chart of the study population and multi-omics data analysis in *Bos indicus* cattle (discovery population)." for further clarification.

Line 207-207 More description is required here. What is the functional relevance of the lead trans eQTLs being located in ATAC-Seq or ChIP peaks? What is the significance of the datasets used e.g. for the purposes of annotation of the bovine genome as a resource to link complex traits to functional genomic regions.

authors: More discussion has been added in line 226-234.

For figure 3 could any of the GWAS hits be annotated?

authors: The results, describing annotating of the GWAS hits, have been reported in Supplementary table 3.

Methods

Including some of this information about the study population earlier in the manuscript would make the rationale for the study and narrative of the results and discussion much clearer.

Could more information in Fig 1 be included showing the different breeds in the datasets used for each part of the analysis?

authors: More information was added to the main text, particularly the introduction and the Fig 1 caption.

Were the red blood cells lysed before the RNA extractions were performed? Would the expression profiles be predominantly associated with white blood cells?

authors: Yes, The RNAProtect actually lyses all blood cells and stabilises the RNA. So the gene expression was a reflection of all blood cell types (whole blood).

Line 411 After filtering for expression ‘of the 10,455 genes included in this analysis’ were these genes particularly associated with any functional roles e.g. immunity?

authors: Thank you for your question. Below is the Gene ontology graph depicting the enrichment of top 20 most significant terms, molecular function (red) and cellular component (blue), among the tested genes. The number of genes associated with each ontology term is indicated in front of the respective bar. The graph has been added to the Supplementary File2.

The datasets haven't been deposited in the public repositories yet but the authors indicate where the accession will be added in the manuscript. This should ideally be done before the manuscript is resubmitted.

authors: The project accession was added in the manuscript.

Specific line changes

Line 50 change 'demonstrates' to 'demonstrating'

authors: It was edited.

Line 82 Add 'in humans' after 'menarche'

authors: It was edited.

Line 160-161 Change to 'The current results provide'

authors: It was edited.

Line 183-184 Include whether this study is on cattle.

authors: It was edited.

Line 184-185 and presumably also sex and age of animals?

authors: It was included.

Line 190-192 Provide details of the genome and annotation used to define these regions.

authors: It was included.

Line 193-194 Add citation or accession and some description of the ChIP data, which tissue etc?

authors: citation was added.

Line 206 Add 'a' before 'previous' and add whether the study is on cattle.

authors: It was edited.

Line 218 Delete 'variants?'

authors: It was edited.

Line 220 On the expression data from whole blood?

authors: It was edited.

Line 230 Add reference for the previous study.

authors: It was edited.

Line 237 Change 'the' to 'a'

authors: It was edited.

Line 241 Add that the effect might be more obvious in other tissues, but is difficult to sample other tissues at scale in the same way as blood.

authors: It was edited.

Line 275 Remove the extra 'the'

authors: It was edited.

Line 290 I can't see how fetal age in weeks is a heritable trait as it is explained here?

authors: Fetal age in weeks, which is recorded via manual palpation at pregnancy diagnosis, could be used as a proxy for heifer puberty. Foetal age in weeks reflects how quickly after bulls enter the paddock that the females conceive. It includes component traits such as how early the females cycle after the bulls enter the paddock. These traits certainly have a genetic component.

Line 392-393 How were the blood samples stored and preserved prior to RNA extraction?

authors: After incubation, the tubes were transported at -20°C, then stored at -80°C until extraction. The details have been added into the text.

Line 400 What was the sequencing depth per sample?

authors: Mean depth (after sequence quality control) was 46,618,531 reads, with a minimum of 15,424,975 and a maximum of 68,193,182. More information has been added to the main text.

Line 452 Please include the ENA accessions for these datasets and also add the cattle gene expression atlas reference here for the data shown in Fig 4.

authors: The ENA accessions for these datasets have been included in the main text. The citation for the cattle gene expression atlas has been included.

Reviewer #2 (Remarks to the Author):

Forutan et al. studied the genetic underpinning of female fertility, an economically important yet challenging trait in cattle. They use multiple proxy phenotypes in two cattle populations and integrate RNA seq data to identify 87 genes putatively associated with female fertility. They further show evidence for a shared pool of genes regulating fertility in cattle and human. The paper is thus of interest to the broad readership of the journal.

The paper is interesting and generally well written. However, I have the following concerns

1. Authors motivate integration eQTL data (Line 217) by stating significant GWAS loci which are also eQTL are more likely to be causal variants. Is there a reason why splicing QTL were not studied? Given there is limited overlap between GWAS and eQTL signals (Line 237), it might be worth looking into splicing QTL.

authors: This is a good point and we are studying splicing QTL, the results will be published in an upcoming paper.

2. It is not clear in the result section that the trans-eQTL scan was limited to variants significant in GWA and cis-eQTL mapping analyses. A discussion on how many cis-eQTL showed trans effect is perhaps interesting.

authors: We observed 148 eQTLs having a trans effect on at least 2 genes contributing to 338 trans-eQTLs. Interestingly, 755 eQTLs exhibited both cis and trans effects, with 448 of them exerting a cis

effect on at least 2 genes, contributing to a total of 1870 *cis*-eQTLs. This information has been added to the manuscript.

3. The proportion of variance explained by the 255 associated variants is reported only for one trait (heifer puberty). A discussion on variance explained for the other four fertility traits could be interesting.

authors: This is a good idea, however in our validation population we have only one fertility trait, *corpus luteum* score (**CL600**), in a multi-breed population. A proper validation for the four traits in the discovery population would require splitting our already small discovery population into discovery and validation (otherwise variance explained will be over-estimated as we discover the variants in the discovery population and then estimate the variance explained in the same population), and this could lead to very few significant results.

Few other minor points

1. In methods, (example #358) heifer puberty is referred to as CL600.

authors: Yes, this is correct. We have renamed this trait “heifer puberty” to aid uptake of GEBV for this trait by Australian cattle producers, as CL600 did not resonate with them (e.g. Hayes et al. 2023).

2. Line #80: “some overlap in” – perhaps authors could provide numbers here.

authors: We now include the number of genes shared between cattle and humans in the text, as suggested.

3. Line #124: Not clear what do authors mean by “full GWAS”.

authors: The word “full” was deleted.

Reviewer #3 (Remarks to the Author):

The manuscript reports identification of genes affecting fertility in cattle. Fertility traits are economically important to beef cattle production. The authors conducted a comprehensive study and the results are very valuable to both the science community and the industry. In general, the experimental design and data analyses were well described and the results were well presented and discussed. The manuscript can be accepted for publication after addressing the following comments:

Line 154, “these loci explained 2.3% of the variation”. Was it phenotypic or genetic variance? Please clarify in the manuscript.

authors: That is the proportion of phenotypic variance explained by the significant SNPs. We have clarified this in the manuscript.

Line 187. “Seventy-eight percent of the lead *cis*-eQTLs were close to the respective gene start site”. How many *cis*-eQTL were within the respective gene? i.e. a DNA variant within a gene was found to be associated with the gene expression of the same gene.

authors: Thank you for bringing this to our attention. Around 22% of lead *cis*-eQTLs (951 out of 4376) were located within the respective gene. We have added this to the result section.

Line 311 “Only 29 autosomes and variants with imputation accuracy of greater than 0.4 and MAF>0.01 (31,140,417) were used for GWAS study”. An imputation accuracy of 0.4 seems very low (also in line 413). Please justify why an accuracy of greater than 0.4 was used. In addition, please briefly describe how the imputation accuracy of DNA variants was assessed.

authors: The imputation accuracy was assessed by estimating the correlation squared of true and imputed genotypes (see Supplementary Figure 2). Choosing an appropriate imputation accuracy is important as it is a balance between not excluding variants with no information and retaining rare

variants. So, we removed variants with imputation accuracy less than 0.4, as suggested by Pistis, Giorgio, et al. "Rare variant genotype imputation with thousands of study-specific whole-genome sequences: implications for cost-effective study designs." *European Journal of Human Genetics* 23.7 (2015): 975-983., where those authors conducted extensive investigations on the impact of different accuracy thresholds on GWAS results. The reference has been added to the manuscript.

Line 324, please indicate how the genotypes were coded and what variant effect (i.e allele substitution or additive effect) was estimated? Based on supplementary tables, it looks like that allele substitution effect (b) was estimated. But please include the SNP effect estimation information in the manuscript.

authors: The genotypes at each SNP locus were coded as 0, 1, or 2, representing the number of copies of the reference allele carried by an individual. The additive effect (fixed effect) of the candidate SNP to be tested for association was estimated using GCTA, after centering and scaling of genotypes according to Yang et al. (2010). Further clarification was included in the main text within the Methods section (Genome-Wide Association Analysis subsection).

Line 327, Please describe how the binary trait “Heifer pregnancy status” was analyzed.

authors: The choice of fixed covariate effects was determined using the `glm` function in R. Subsequently, we employed a mixed linear model in GCTA to analyse the binary trait of heifer pregnancy status. The frequency of Heifer pregnancy was 0.74, and previous investigations have shown that provided that frequency is between 0.2 and 0.8, linear models are a fair (though not perfect of course) approximation of binary models for binary traits (also considering the computational feasibility of analysing so many variants with these models).

Line 328, please describe how was the contemporary groups defined, and how many levels of the contemporary groups.

authors: The contemporary group was defined as -season-year effect, with 7 levels. It has been added to the manuscript.

Line 330, “for trait AFC, calving success defined as 0 and 1, based upon whether they calved before or after 900 days of age was considered as a fixed effect”. Does this mean if the cow calved before 900 days of age, its calving success was assigned “1”, or otherwise as “0”. Please justify why calving success was considered as a fixed effect for AFC.

authors: We observed a bimodal distribution of AFC, with 2 peaks depending on whether calving happened before or after 900 days. So, we considered calving success as a fixed effect to account for its effect on AFC. Here are the QQ plot of GWAS P-values before (left) and after (right) fitting calving success as fixed effect into model.

Line 366, please describe phenotype values of the validation population in more detail. Were they pre-adjusted for non-genetic effects?

authors: The effect of contemporary group and *Bos indicus* content was fitted as fixed effects in

prediction model. The main text has been revised.

Line 380. Did the 489 *Bos indicus* heifers and cows used for expression analysis have the same or similar four fertility phenotypes? If yes, did the 489 cattle have similar mean and ranges of variation as the population used for the GWAS?

authors: Thanks for your valuable comments. To provide insights into the comparability and representativeness of the sample used for the eQTL analysis in relation to the discovery population studied in GWAS, a table describing the summary statistics, including range, mean, and standard deviation (SD), for the animals from the eQTL study as well as discovery population has been added in the supplementary file (Supplementary File 1: Table S1). And yes, these were similar to the discovery population.

Table S1, it would be more informative to add mean, range of phenotypic values, SD of the four traits and their estimates of genomic Heritability from the GWAS?

authors: As suggested a table describing the summary statistics, including range, mean, and standard deviation (SD), and heritability has been added in the supplementary file (Supplementary File 1: Table S1).

Table S2 and Table S4, It would be more informative to add SNP annotation (A1, A2) and indicate which SNP allele is the minor allele with the allele frequency, i.e. similar to Table S6 on SNP information (A1 A2).

authors: The two tables have been updated as suggested.

REVIEWERS' COMMENTS:

Reviewer #1 (Remarks to the Author):

The authors have done an excellent job of comprehensively addressing all of my comments and uploaded the datasets to the public repositories to generate an accession number as requested. I believe this manuscript can now be accepted for publication.

I just have two small suggestions 1) edit line 233 to remove "beyond mere genome annotation" which is a bit counterintuitive based on the rest of the sentence as it's currently written. 2) would it be possible to add a definition of the trait 'fetal age at calving' which explains why it has a genetic component. The authors explain this well in their response to reviewers but it would be helpful to also add it to the manuscript text to avoid the confusion I had as someone not familiar with how some traits are recorded in cattle.

Reviewer #2 (Remarks to the Author):

Authors have addressed all my comments in the rebuttal letter and have also modified the manuscript. I have no further comments